# Health status of street children and reasons for being forced to live on the streets in Harar, Eastern Ethiopia. Using mixed methods

**Degu Abate[1], Addis Eyeberu[2]\*, Dechasa Adare[3], Belay Negash[4], Addisu Alemu[4], Temam Beshir[4], Alemayehu Deressa Wayessa[4], Adera Debella[2], Nebiyu Bahiru[4], Helina Heluf[2], Mohammed Abdurke Kure[2], Ahmedmenewer Abdu[1], Amanuel Oljira Dulo[2], Habtamu Bekele[2], Kefelegn Bayu[3], Saron Bogale[5], Genanaw Atnafe[2], Tewodros Assefa[5], Rabuma Belete[1], Mohammed Muzeyin[3], Haftu Asmerom[1], Mesay Arkew[1], Anumein Mohammed[1], Henock Asfaw[2], Barkot Taddesse[2], Daniel Alemu[2], Dawit Yihun[5], Shambel Nigussie[5], Jemal Yusuf Kebira[4], Siraj Aliyi Adem[2], Gebisa Dirirsa[3], Saba Hailu[4], Abduro Godana[5], Galana Mamo[4], Deribe Bekele[2], Yadeta Dessie[4]**

1 Department of Medical Laboratory Sciences, College of Health and Medical Sciences, Haramaya University, Harar, Ethiopia, 2 School of Nursing and Midwifery, College of Health and Medical Sciences, Haramaya University, Harar, Ethiopia, 3 Department of Environmental Health, College of Health and Medical Science, Haramaya University, Harar, Ethiopia, 4 School of Public Health, College of Health and Medical Sciences, Haramaya University, Harar, Ethiopia, 5 School of Pharmacy, College of Health and Medical Sciences, Haramaya University, Harar, Ethiopia

* addiseyeberu@gmail.com

**Data Availability Statement:** All relevant data are within the manuscript and its Supporting Information files.

## Abstract

### Introduction

In Ethiopia, more than four million children are anticipated to live under particularly difficult circumstances. Street children are subject to violence, a lack of health care, and a lack of education. Which denies them the right to live in a secure environment and exposes them to different health problems. Currently, little is known about the prevalence of Streetism, including health conditions. Therefore, this study was aimed to assess the health status of street children and determinants of Streetism.

### Methods

Mixed methods (sequential) were employed from February 1 to 28, 2021. Quantitative cross-sectional study design and phenomenological qualitative designs were applied. Over-all, 220 street children were involved in the study. The most common reason that forced the children to resort to a street way of life is to look for a job and quarreled with parents. The data were collected using interviews methods. Chi-square test and multiple binary logistic regression were applied to examine the variations among variables with the health status of street children. Qualitative data were analyzed using the thematic analysis technique.

### Results

The study included a total of 220 street children. As to the quantitative study, the majority of study participants (92.73%) drank alcohol regularly. Depression (39.22%) and peer

**Funding:** The authors received no specific funding for this work.

**Competing interests:** The authors have declared that no competing interests exist.

pressure (43.14%) were the most common initiation causes of drinking alcohol. According to a qualitative study report, "Street children are mostly affected by the communicable disease" and. . . They are addicted to substances like benzene" which had a profound effect on their health." Furthermore, the study discovered a statistically significant association between respondents' health status and sociodemographic characteristics (age and educational status), job presence, and drug use.

## Conclusion

This study identified the factors that drove street children to live on the streets, such as the inability to find work and disagreements with their parents. The majority of the street children were affected by preventable and treatable diseases. Unfortunately, almost all street children reported drinking alcohol, which exposed them to a variety of health problems. In general, the study discovered that street children require immediate attention. Decision-makers and academicians should collaborate to develop a plan for these children's health and social interventions.

## Introduction

The United Nations Children's Fund (UNICEF) defines street children as "children in difficult circumstances," who are a minority population that has been under-represented in health research for far too long. The street child population has also been divided into two overlapping groups: "of-the-street" children, who have no contact with family and rarely return home, and "on-the-street" children, who often sleep at home but spend the day on the street [1, 2].

According to studies, among the major health issues affecting street children are physical injury, HIV/AIDS, sexually transmitted diseases, sexual and reproductive health disorders, violence and sexual abuse, substance abuse, and mental health problems. They are considered the most vulnerable population; they are severely impoverished and lack access to health care and education. They are also victims of ferocity and are forced to live on the streets, scavenging, begging, hawking in slums, and living in polluted environments [1, 3, 4]. Street children are subjected to violence; denying them the right to live in a safe environment [5–7].

According to the study, children who spend time on the street are at risk of developing health and social problems, as well as bizarre behaviors such as aggression and hopelessness [8]. Due to hunger, malnutrition, and hygiene issues, street children face a variety of issues that have the potential to negatively impact their health [9]. Evidence showed that the most common reasons forced street children to live in the street were escaping abusive parental punishment, poverty, and parental alcoholic behavior [10]. A similar report also showed that deaths of parents and unhealthy relationships of families were causes that force street children to stay on the streets [11].

Worldwide, around 28 million children live in the street, while in Ethiopia's recent report by the Ministry of Labour and Social Affairs, the number of homeless people in Addis Ababa was around 24,000 in 2018; approximately 10,500 street children and 13,500 homeless adults [12]. Similar evidence showed that in Ethiopia, over four million children are anticipated to live under particularly difficult circumstances [13–15]. They are at high risk of sexual and physical exploitation [16]. Evidence showed that 15.6% of the street children are practicing risky sexual activity, and 61.6% of the street children face health problems [17]. Nonetheless,

despite the growing burden of health problems among Ethiopia's street children, there is no policy emphasis on the country's health system.

It is now being observed that the number of street children is significantly increasing; however, little is known about the prevalence of the problems, including the factors that lead to being a street child and their health status [1, 18]. There is a lack of comprehensive and adequate information about street children to take action, necessitating the current study to investigate the street children's health status and factors leading to being street children in Harar, Eastern Ethiopia. As a result, the objective of this study was to assess the health status of street children in Harari City, as well as to explore the factors that contribute to being street children, to assist policymakers, administrators, health program planners, and researchers in finding solutions to the problems.

## Materials and methods

### Study setting and period

The study was carried out in Harari regional state, Eastern Ethiopia. Harar is located 522 kilometers east of Addis Ababa, Ethiopia's capital, at an elevation of 1,885 meters above sea level. The region has a total population of 183,344 people, including 84047 rural residents and 99368 urban residents. Since 2006, UNESCO has designated Harar as a World Cultural Heritage Site. Currently, the city is one of the fastest-growing in Ethiopia, attracting many adults and children from the surrounding rural areas and other neighboring cities [19].

### Study design and population

Sequential mixed study designs (quantitative and qualitative) were applied. First collection and analysis of quantitative data followed by a collection and analysis of qualitative data was done. The qualitative results was used to assist in explaining and interpreting the findings of a quantitative study. This was done according to Creswell, 2003 evidence of using mixed-methods sequential explanatory design. Quantitative cross-sectional study design and phenomenological qualitative designs were applied. The source population was all street children living in Harar city while all selected street children residing in Harar city during the study period were the study populations. Study participants that are unable to respond to the survey questionnaire during the data collection period were excluded.

### Sample size determination and sampling procedure

Street children can be found in a variety of locations throughout Harar. Seven are chosen on purpose because there is a higher concentration of street children in these areas of town known as Arategna, Ajip, Canal, Botte, Shewa ber, Feres Megala, and Andegan Menged. This was accomplished through observation and consultation with key informants working with street children. Because there was no street children list in Harar city's Social Affairs Office, the researchers compiled a comprehensive, reliable, and appropriate list by registering all street children living or working on the streets. The registration of street children was completed five days before the data collection date. As a result, 502 street children were registered in the seven sub-towns (S1 Table).

The sample size for quantitative data was calculated using the single population proportion formula (n = (Za/2)2p(1-p)/d 2), with the assumptions of Z = 1.96 for the 95% confidence interval, d = 0.05 for margin of error, and p = 0.5 for the prevalence of Streetism, yielding n = (1.96)2(0.5) (1–0.5) / (0.05)2, n = (3.8416) x0.5 (0.5) /0.0025, and since the total population was less than 10,000, we used the reduction formal, and the final sample size was 228.

For qualitative data, the sample size was determined based on the level of information saturation and the variety of ideas among the sub-groups. Data saturation involves sampling until no new information was obtained and redundancy was achieved. Accordingly, twelve focus group discussions (FGDs) with 8 participants from various groups and 6 key informants were carried out.

Participants include experts from the Women and Children Affairs Office, Police Office, Social Affairs Office, local NGOs, street children's families, and street children were carried out. These key informants and focus group discussants were selected purposively since they are concerned and more informed bodies in the area. Data collectors have availed themselves in the selected area of the studies in scheduled and pre-planned time to include all street children available in the sites consecutively till the calculated sample size was achieved and for qualitative data, key informants and focus group discussant includes experts from Women and Children Affairs Office, Police Office, Social Affairs Office, local NGOs, street children families and street children as they were more informed and concerned bodies in the area of interest.

## Data collection methods and data collection procedures

To fit the local situation and research objectives, the adapted questionnaires were modified and contextualized. For quantitative data, interviewer-administered questionnaires were used. We tried to do our best to adapt a tool named "the health status of street children survey" from measure evaluation which was funded by USAID and undertaken with the title of "children in adverse situations indicators and survey tools and previous literature (BSS tool). The questionnaire includes socio-demographic characteristics of the study participants, life conditions and reasons of Streetism of the study participants, and the health status of street children.

The questionaries were validated before administering to study participants. The first steps were translating the English version questionaries to the local language in the study area (Afan Oromo and Amharic) while keeping the purpose of the questionnaire and the intent of the questions in mind and again translated back to the English language to check the consistency and to ensure the translation's accuracy. The back-translated version is then compared to the original, and any meaning differences are corrected. The second step was pretesting on the 5% of the respondents who were not eligible for the study before actual data collection. Then the validity and reliability of the instrument were checked. Cronbach's Alpha was used to measure the quality of our employed instruments. The result was 0.87. Six data collectors and two supervisors were recruited who could communicate in the local language of the study area and preferably had experience in other similar field studies. Every day, the collected data were checked for completeness and consistency. Data collectors and supervisors were also trained. Key informant interviews were used to collect qualitative. A semi-structured interview guide was used. Each key informant's data was recorded in an audio recorder in a controlled setting to avoid disruption during the interview.

## Data quality control and assurance

The investigators and supervisors were made a thorough check for clarity, completeness, and consistency before receiving the filled questionnaire from each data collector. The quality of the data was guaranteed by pretesting using the 5% of the respondents who were not eligible for the study before actual data collection. During the collection of data, questionnaires were cleaned and properly coded.

To assure the qualitative data quality, the key informant interview was recorded using an audio recorder. Member check for the interviews (after the transcription, by summarizing

main points and confirming with the interviewee). The initial results were shown for the peers to receive input (Peer debriefs). Moreover, triangulation through different key informant interviews was employed.

## Statical analysis

The data were coded, cleaned, edited, and entered into EPI data version 3.1 to minimize logical errors and design skipping patterns. Then, the data were exported to Stata version 14 for analysis. descriptive analysis was done by computing proportions and summary statistics. Then, the information was presented by using simple frequencies, summary measures, and tables. Multi-variable analysis was carried out to see the association between each independent variable and outcome variables by using binary logistic regression. The assumption for binary logistic regression was cheeked. The goodness of fit was checked by Hosmer-Lemeshow statistic and omnibus tests. All variables with $p<0.25$ in the Bivariate analysis were included in the final model of multivariate analysis to control all possible confounders. A Multi-collinearity test was carried out to see the correlation between independent variables by using the standard error and collinearity statistics (no variable with variance inflation factors $>10$ and standard error $>2$ was observed). The direction and strength statistical association was measured by odds ratio with 95% Confidence Interval (CI). Adjusted odds ratio along with 95%CI was estimated to identify predictors of the health status of the streets by using multivariate analysis in binary logistic regression. In this study, P-value$<0.05$ was considered to declare a result statistically significant.

Qualitative data were analyzed using Atlas Ti version 7.1. The thematic analysis technique was used. The recorded data were transcribed and translated. Then the translated data were coded into different codes. Each code was categorized into different categories and then categorized into themes.

## Ethical consideration

Ethical clearance was secured from Haramaya University, College of Health and Medical Sciences, Institutional Health Research Ethical Review Committee (IHRERC). The approval number was IHRERC/016/2021. Before data collection was started, informed, voluntary, written and signed consent was taken from the respondents. For minors (under the age of 18), consent was obtained from their parents and guardians. But for those individuals, without parents, we took assents from children themselves. The institutional health research ethical review committee approved obtaining consent from children that did not have parents or guardians. Any study participant willing to engage in the study and those who want to stop the interview at any time were allowed to do so. Confidentiality was also assured. All the procedures are done per the ethical guidelines of the institution.

## Results

### Socio-demographic characteristics

A total of 220 respondents participated in the study yielding a response rate of 96.5%. The median age of study participants was 14 years, with an Inter-Quartile range of 4 years and minimum and maximum ages were 5 and 44 years, respectively. Of the total respondents, 78.18% were males. The majority of the participants were Muslims (89.55%). Twenty-eight (12.73%) of the study participants were married. The average monthly income of their parents was 1526 Ethiopian Birrs while the street children got 90.2 Ethiopian Birr on average (Table 1).

**Table 1. Socio-demographic characteristics of the study participants and their parents in Harar, Eastern Ethiopia, 2021 (n = 220).**

| Variables | Category | Frequency | Percentage |
|---|---|---|---|
| Age | below12 years | 75 | 34.09 |
| | 13–18 years | 120 | 54.55 |
| | Above 19 years | 25 | 11.36 |
| Sex | Male | 172 | 78.18 |
| | Female | 48 | 21.82 |
| Address before Streetism | Urban | 119 | 54.1 |
| | Rural | 101 | 45.9 |
| Religion | Muslim | 197 | 89.55 |
| | Orthodox | 18 | 8.18 |
| | Protestant | 5 | 2.27 |
| Ethnicity | Oromo | 200 | 90.91 |
| | Amhara | 12 | 5.45 |
| | Hadere | 2 | 0.9 |
| | *Others | 6 | 2.73 |
| Marital status | Never married/ not eligible | 185 | 84.09 |
| | Married | 28 | 12.73 |
| | Widowed | 1 | 0.45 |
| | Divorced | 7 | 3.18 |
| Fathers' educational status | Illiterate | 146 | 66.36 |
| | Can read and write | 27 | 12.27 |
| | Primary and secondary grades completed | 47 | 21.36 |
| Mother's means of livelihood | Informal daily laborer | 63 | 28.64 |
| | petty seller/trader | 42 | 19.09 |
| | skilled worker/self-employed | 3 | 1.36 |
| | house servant | 29 | 13.18 |
| | Beggar | 12 | 5.45 |
| | house wife | 39 | 17.73 |
| | **Others | 32 | 14.55 |
| Father's means of income | Farmer | 112 | 50.91 |
| | Informal daily laborer | 48 | 21.82 |
| | Trader | 8 | 3.64 |
| | Government employee | 4 | 1.82 |
| | Private employee | 5 | 2.27 |
| | Beggar | 5 | 2.27 |
| | Soldier/x-soldier | 7 | 3.18 |
| | Guard | 7 | 3.18 |
| | ***Other | 24 | 10.91 |
| Where parents live know (n = 201) | In Harar | 62 | 30.85 |
| | Another urban city | 35 | 17.41 |
| | Rural area | 101 | 50.25 |
| | Don't know | 3 | 1.49 |
| The reason they leave their place of origin (if parents were migrants) | Drought/famine situation | 27 | 27 |
| | for medical treatment | 16 | 16.00 |
| | To look for a job | 34 | 34 |
| | Don't know | 23 | 23 |

(*Continued*)

**Table 1.** (Continued)

| Variables | Category | Frequency | Percentage |
|---|---|---|---|
| Income level of parents | Less than 1000 | 163 | 74.09 |
| | 1000–2000 | 30 | 13.64 |
| | Above 2000 | 27 | 12.27 |

* Others = Gurage, Sidamo, and Somalia

**Others = no work, prostitute, and don't know

***Other = don't know and no work.

## Educational status of study participants

More than half (67.73%) of the study participants attained primary education(1-4grade) and forty (18.18%) of the street children were never enrolled in the school. The most common reasons hindering them from attending the school include family poverty/financial constraints and family did not appreciate education (Table 2).

## Prevalence of divorce among parents of street children

The magnitude of divorce among parents of street children was 24.63% [95%CI 18.2, 31.8]. The most common causes of divorce were bad habits 10 (31.25%) and disagreement/ conflict among parents 14 (43.75%). The other causes of divorce include poverty 7 (21.88%) and father leaving home to look for a job 1(13.13%). The majority (91.36%) of the streets have parents

**Table 2.** Educational status of the study participants in Harar, Eastern Ethiopia, 2021.

| Variables | Category | Frequency | Percentage |
|---|---|---|---|
| Educational status (n = 220) | Never attain school | 40 | 18.18 |
| | Read and write | 18 | 8.18 |
| | 1–4 grade | 149 | 67.73 |
| | 5–8 grade | 11 | 5.00 |
| | 9–12 grade | 2 | 0.91 |
| Factors hindered from attending school (n = 40) | Family poverty/Financial constraints | 28 | 70 |
| | The family did not attach value to education | 6 | 15 |
| | Had to remain home to help the family | 4 | 10 |
| | Others (bad conduct, too small for education) | 2 | 5 |
| Reason for school dropout (n = 79) | The family could no longer afford the school fee | 21 | 26.58 |
| | I had to work to supplement the family income | 21 | 26.58 |
| | Health/medical reasons | 1 | 1.27 |
| | Expelled because of conduct | 10 | 12.66 |
| | To make my reading | 2 | 2.53 |
| | *Others | 24 | 30.38 |
| If you are given the chance to continue your education now, are you willing to and happy to make use of the opportunity? (n = 79) | Yes | 72 | 91.14 |
| | No | 7 | 8.86 |

*Others = conflict with family and no reason.

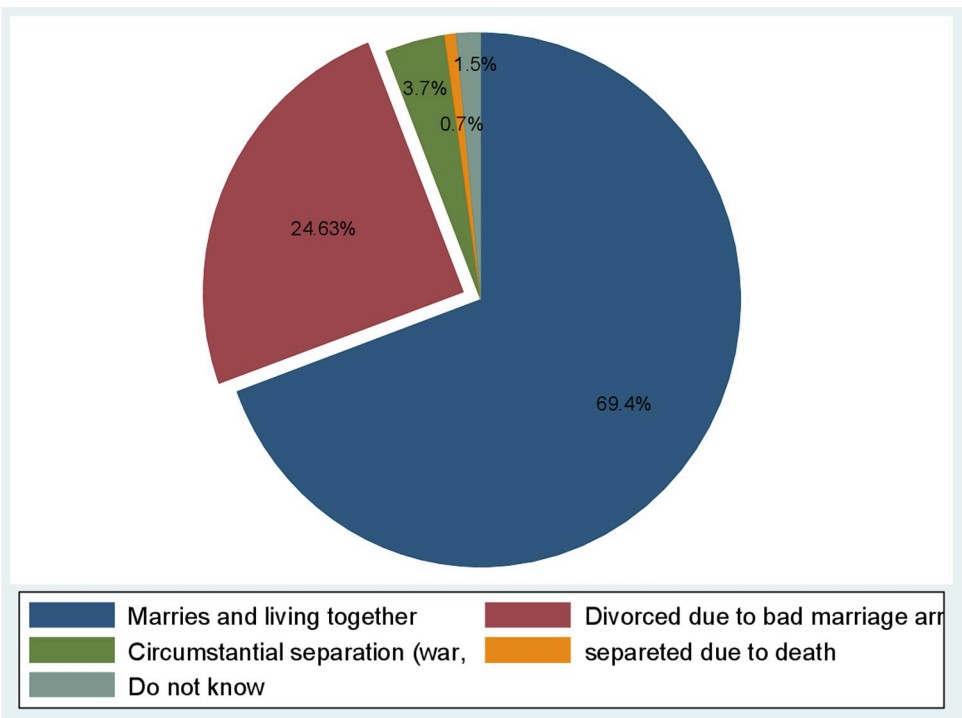

**Fig 1. Prevalence of divorce among parents of street children Harar, eastern Ethiopia, 2021.**

and about 19(8.64%) children did not have parents. From those who reported they have parents, more than half 134 (66.67%) of the street children's parents were alive while 48 (23.88%) of street children had only mothers alive (Fig 1).

## Reasons for being forced to live in the street and their life conditions

**Quantitative result.** This study revealed that the mean length of stay on the street was 3.13 (SD = 3.45) years. Where more than half (55%) of the study participants have lived on the street for less than 1 year. The most common reasons that forced them to resort to the street way of life include looking for a job (26.6%) and quarreling with parents (26.6%). Furthermore, peer pressure (15%), the need for food (8.64%), and the death of parents (5%) were other factors that forced street children to restore ways of life (Fig 2).

**Qualitative findings.** As it was clearly shown in Table 3, the common reason forcing street children to live in the street were organized under two major themes include management and stakeholder (Table 3).

Management related factors.

*"There is inadequate financial support! We have a budget problem. . .Many workers were turned over due to the lack of encouragement! There is no assessment done, to have full information about street children found in Harar! There is lack of transparencies among workers. . ."* (Report from a 32-year-old female working as women and children affair).

Stake Holder's related factors.

*"There are some volunteer centers for rehabilitation named Tesfa, Shalom. . .in Harar but they are insufficient in comparison to the victims. Some help from volunteers is only focused*

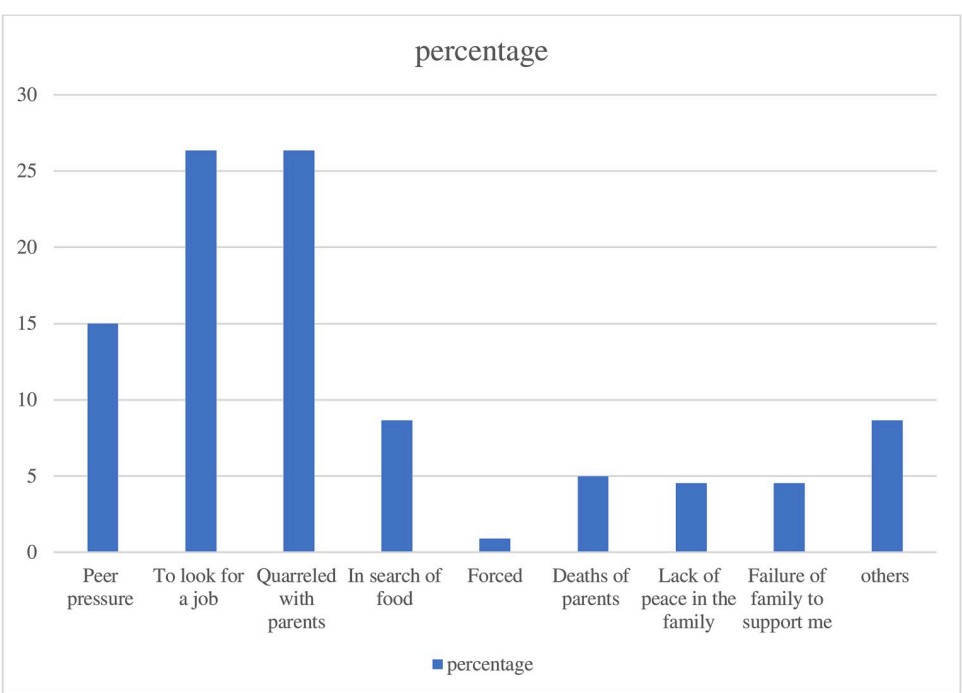

**Fig 2. Factor that forced the street children to live in the street in Harar, Ethiopia, 2021.**

*on something temporary like supporting food and clothes and so forth. But it does not focus on something permanent like rehabilitating and educating... many people are not taking responsibility to solve street children's problem, but everybody should do something! Father-in-law and mother-in-law should also treat every child. There is a lack of attention towards street children among society...society should discourage substance use first... there is inadequate participation among society..." (Report from a 32-year-old female working as women and children affair)*

**Quantitative finding.** This finding showed that more than half (63.64%) of the streets didn't have work. Sixty-four (36.78%) of the streets survive by begging while 45 (25.86%),36 (20.69%) of the streets survive by carrying small items and serving as a taxi boy, respectively. But eighty-nine (40.45%) of the streets beg even if they have other occupations, and 52.81% of those who beg always beg. Almost half (49.43%) of the streets earn less than 50 birrs per day, with three-quarters (76%) spending it on food. Eighty-four (38.18%) of the streets slept on the verandah (Table 4).

## Health status and substance use among street children

**Quantitative results.** The majority (61.36%) of the streets got sick in the past month. One hundred twenty-nine (58.64%) of the study participant were hurt by adults while they live in the street. More than one-fourth (27.27%) of the participants had a history of sexual intercourse with someone. Among those, 14.55% had forced sexual intercourse and 8.64% of them had sexual intercourse for sake of money.

Almost all (92.73%) of the study participants drank alcoholic beverages at different frequencies. The most common initiation causes were depression (80, 39.22%) and peer pressure (88, 43.14%). About 60 (27.7%) of the participants used different kinds of drugs like chat chewing, Ganja/Hashish, cigarette smoking, sniff benzene (Table 5).

**Table 3. Explored reasons forcing street children to live in the street in Harar, eastern Ethiopia, 2021.**

| SN | Theme | Categories | Sub categories | Codes |
|---|---|---|---|---|
| 1 | Management issue | Budgeting | Budget to solve problem | B inadequate financial support |
| | | | | B lack of budget |
| | | | Workers incentive | W many workers were turned over |
| | | | | W workers are discouraging |
| | | Supervision | Assessment | A there was no assessment done yet |
| | | | | A we don't have exact data of s. children |
| | | | Transparency | T communication gap among stakeholders |
| | | | | T lack of transparencies among workers |
| 2 | Stakeholders | Supporting agents | NGO /volunteers agencies | V rehabilitation center but inadequate |
| | | | | V focus on supporting food and cloths only |
| | | | Governmental agencies | G lack of transparency among workers |
| | | | | G lack of taking responsibility to solve a problem |
| | | | | G every one should do something |
| | | Family-related factors | Father/ mother in low | F Loss of family is one factor to live street |
| | | | | F Father and mother in law also the factor |
| | | | Low-income family | I Lack support in food, shelter, cloth, and school |
| | | | | I They complain to have food and money |
| | | | Lack of awareness | Aw lack of awareness on family planning |
| | | | | Aw an illiterate families /carelessness |
| | | Society related | Attitude | A Lack of Attention towards S children |
| | | | | A society should discourage substance use first |
| | | | Participation | P every one should do something but not yet |
| | | | | P there is inadequate participation among society |
| 3 | The lifestyle and health conditions of street children | The behavior of street children's | Unwillingness | U street children disobey their family advice |
| | | | | U unwilling to return to their family |
| | | | Substance use | S Usage of an addictive substance, Use benzenes |
| | | | | S Escaped from rehabilitation centers due to craving |
| | | Health related issue | Sexual violence | S Sexual violence |
| | | | | Unprotected sex |
| | | | | S Childbearing age females mostly use implant |
| | | | Disease | D they use unclean food from dirty tankers |
| | | | | D They mostly affected by communicable disease |
| | | | Accident | Ac Raped |
| | | | | Ac Traffic accident |

**Qualitative results.** Study participants reported that street children were commonly affected by communicable diseases and substance addiction.

*"Street children are mostly affected by communicable disease. . . they use unclean food from dirty tankers. . .they complain to have food, money to buy food. . . Some street children have difficult behavior*! They *escaped from rehabilitation centers due to cravings from substance addiction like benzene. Families complain their child disobeys their advice. Unwilling to return to their family. . .. Child-bearing age females mostly use the implant as a result of fearing sexual violence, unprotected sex-related pregnancy. Childbearing age females mostly use implant. . . of course, there are accidents related trafficking and also raping on the street"* *(report for 45 years old male participant)*

**Table 4. Life condition or circumstances of streets in Harar, Eastern Ethiopia, 2021.**

| Variables | Category | Frequency | Percentage |
|---|---|---|---|
| Do you have work (n = 220) | Yes | 80 | 36.36 |
| | No | 140 | 63.64 |
| street occupation (n = 174) | Begging | 64 | 36.78 |
| | Shoe shinning | 4 | 2.30 |
| | Carrying small items | 45 | 25.86 |
| | Delivering messages | 25 | 14.37 |
| | Taxi boy | 36 | 20.69 |
| Daily income (174) | Less than fifty birrs | 86 | 49.43 |
| | 50–100 birr | 53 | 30.46 |
| | more than 100 birrs | 35 | 20.1 |
| Daily income spent on (174) | Food | 133 | 76.00 |
| | Cloth | 2 | 1.14 |
| | School fee | 38 | 21.71 |
| | Help family | 2 | 1.14 |
| Daily income reliable (n = 174) | Yes | 27 | 15.52 |
| | No | 147 | 84.48 |
| With whom were you living before you resort to street life? | With both parents | 113 | 51.83 |
| | With my mother | 57 | 26.15 |
| | With father only | 17 | 7.80 |
| | With close relatives | 22 | 10.09 |
| | *Others | 11 | 5 |
| To whom are you living now? | With my mother | 26 | 11.98 |
| | with both parents | 8 | 3.69 |
| | With relatives | 9 | 4.09 |
| | With father Alone | 2 | 0.91 |
| | With peers/friends | 152 | 69.09 |
| | **Other | 23 | 10.45 |
| Who helps and protects you while you are on the street? | No one | 95 | 43.18 |
| | Siblings | 5 | 2.27 |
| | Peer groups | 72 | 32.73 |
| | Parents | 21 | 9.55 |
| | Close relatives | 27 | 12.27 |
| How often do you usually see/visit your families? | Everyday | 41 | 18.64 |
| | at least once a week | 19 | 8.64 |
| | Once every month | 12 | 5.45 |
| | only a few times in a year | 18 | 8.18 |
| | Rarely | 26 | 11.82 |
| | Irregularly | 22 | 10.00 |
| | Never | 18 | 8.18 |
| | Don't care | 64 | 29.09 |
| Approve/disapprove of living or working on the street by parents? | Approve | 44 | 24 |
| | Disapprove | 59 | 32.42 |
| | Don't care | 49 | 26.92 |
| | Don't know | 30 | 16.48 |
| How many times a day do you eat? | Once | 42 | 19.09 |
| | Twice | 82 | 37.27 |
| | Three times and above | 96 | 43.64 |

(*Continued*)

**Table 4.** (Continued)

| Variables | Category | Frequency | Percentage |
|---|---|---|---|
| Where do you usually go to sleep? | Home | 26 | 11.82 |
| | church yards | 1 | 0.45 |
| | mosque yards | 4 | 1.82 |
| | Bus stops | 13 | 5.91 |
| | on verandah | 84 | 38.18 |
| | Plastic houses | 92 | 41.82 |

*Others = friends, wife and husband

**Other = alone and husband.

**Lived experience of street children.** This study pointed out that drug and alcohol abuse is wide spread among street children. One girl stated:

"*To cope, I smoke cigarettes, which I believe helps me deal with the stress of living on the streets. My friends and I enjoy drinking alcohol and smoking cigarettes*".

This participant's actions appeared to be motivated by a desire to enjoy their freedom on the street. The girls in the current study found drinking alcohol and smoking cigarettes, which may have served as a form of escapism from harsh street conditions.

*In a similar vein, one ten-year-old girl stated, "I enjoyed "smoking glue," which generates pleasant feelings while also blocking out hunger, cold, and insecurity."*

Even more so, a 12-year-old male participant who had recently begun life on the streets spoke of sniffing glue and how it helped him by allowing him to escape the stressors of the street:

"*You smoke glue if you don't want to keep thinking about your situation because it gets you high and makes you hallucinate; you don't have to keep thinking about living on the streets and all that stuff. I don't blame anyone who snorts glue; glue alleviates the sadness of living on the streets*".

According to the above narratives, these participants' lives appear to revolve around drugs. As a result, they exhibit symptoms of substance-related disorders in one way or another.

## Factors associated with the health status of the street children

Multivariable logistic regression was done to identify the factors associated with the street children's health status. With that, the age of the respondents was significantly associated with the health status of the streets. Respondents with the age below 12 and 13–18 were 3.43 times [AOR = 3.43, 95%CI (1.13,10.42)], 3.8 times [AOR = 3.80, 95%CI (1.14,10.47)] more likely to become sick when compared to those who are above the age of 19 years old respectively. Fathers' educational status was also significantly associated with the health conditions of the streets. Participants' parents who had no education/were illiterate were 0.37 times [AOR = 0.37,95%CI (0.17,0.84)] less likely to become sick compared to those respondents' parents who have higher grades completed.

**Table 5. Health status and substance use among study participants in Harar city, Ethiopia, 2021.**

| Variables | Category | Frequency | Percentage |
|---|---|---|---|
| Have to gotten sick in the past month while living or working on the streets? | Yes | 135 | 61.36 |
| | No | 85 | 38.64 |
| Have you ever been beaten or hurt by adults while living/working on the streets? | Yes | 129 | 58.64 |
| | No | 91 | 41.36 |
| Have you ever had any sexual intercourse with someone? | Yes | 60 | 27.27 |
| | No | 160 | 72.73 |
| Has anyone forced you to do any sexual act? | Yes | 32 | 14.55 |
| | No | 188 | 85.45 |
| Have you ever had any sexual intercourse with someone for the sake of money? | Yes | 19 | 8.64 |
| | No | 201 | 91.36 |
| Which of the following is your immediate need? | Food | 76 | 34.55 |
| | Cloth | 10 | 4.55 |
| | Shelter | 16 | 7.27 |
| | Education | 37 | 16.82 |
| | Family support | 14 | 6.36 |
| | Cash assistance | 42 | 19.09 |
| | Get reunified | 19 | 8.64 |
| | Job | 6 | 2.73 |
| Do you drink an alcoholic beverage like Tela, Tej, Beer, Arekie, and the likes? | Have never drunk | 16 | 7.27 |
| | I have tried once or twice | 193 | 87.73 |
| | I drink most of the time | 7 | 3.18 |
| | I drink daily | 4 | 1.82 |
| What initiates you to use alcohol? | Depression | 80 | 39.22 |
| | Peer pressure | 88 | 43.14 |
| | To protect hunger | 20 | 9.80 |
| | To protect fear (sex, steal) | 16 | 7.84 |
| Do you use drugs? | Yes | 60 | 27.27 |
| | No | 160 | 72.73 |
| If yes, which drug do you use most of the time? | Chat chewing | 26 | 43.33 |
| | Ganja/Hashish, | 5 | 8.33 |
| | Cigarette smoking | 12 | 20.00 |
| | Sniff Benzene | 14 | 23.33 |
| | Mastish | 3 | 5.00 |
| What initiates you to use drugs? | Depression | 13 | 21.67 |
| | Peer pressure | 31 | 51.67 |
| | To protect hunger | 10 | 16.67 |
| | To protect fear | 1 | 1.67 |
| | *Other | 5 | 8.33 |

*Other to cope with cold weather and to get strength

Having work was significantly associated with the health condition of the streets. The odds of being sick among participants who hadn't worked was 2.12 times that of a participant who had worked [AOR = 2.12, 95%CI (1.12,4.01)].

Similarly, the history of sexual intercourse was also associated with the health conditions of the streets. The odds of being sick among participants who had a history of sexual intercourse was 2.39 times that of a participant who had no history of sexual intercourse [AOR = 2.39, 95% CI (1.06,5.42)] (Table 6).

Table 6. Factors associated with the health status of the streets in Harar, Ethiopia (n = 220).

| Variables | | Health status | | |
|---|---|---|---|---|
| | | Sicked | not sicked | AOR (95%CI) |
| Age | Below 12 years | 45 | 30 | 3.43(1.13,10.42) * |
| | 13–18 years | 80 | 40 | 3.8(1.14,10.47) ** |
| | Above 19 years | 10 | 15 | 1 |
| Gender | Male | 106 | 66 | 1 |
| | Female | 29 | 19 | 1.12(0.52,2.45) |
| Fathers' education | Illiterate | 82 | 64 | 0.37(0.17,0.84) * |
| | Read and write | 17 | 10 | 0.58(0.19,1.75) |
| | Highest grade com. | 12 | 35 | 1 |
| Parents Income | ≤ 1,000 ETB | 100 | 63 | 0.84(0.33,2.14) |
| | 1,001–2,000 ETB | 18 | 12 | 0.76(0.24,2.36) |
| | >2,001 ETB | 17 | 10 | 1 |
| Marital status | Married | 55 | 38 | 1 |
| | Widowed/divorced | 22 | 10 | 1.68(0.65, 4.32) |
| | Not married/died | 37 | 58 | 1.23(0.63,2.39) |
| Drug use | No | 98 | 62 | 1 |
| | Yes | 37 | 23 | 0.96(0.48,1.89) |
| Work presence | Yes | 56 | 24 | 1 |
| | No | 79 | 61 | 2.12(1.12,4.01) ** |
| Sexual history | No | 41 | 19 | 1 |
| | Yes | 94 | 66 | 2.39(1.06,5.42) * |

** = significant at <0.01 and

* = significant at ≤0.05, ETB = Ethiopian Birr

## Discussion

This study assessed the health status of street children as well as the factors that drove them to live on the streets in Harar, eastern Ethiopia. Furthermore, factors affecting the study participant's health status were identified.

In this study, less than one-third (18.8%) of the street children were never enrolled in the school. This proportion is lower than that found in an Indian study (66.5%) [20]. The disparity could be attributed to socio-demographic and cultural differences between study areas. Every child is entitled to an education. Even though the Ethiopian constitution guarantees the right of the child to education, some street children do not receive an education due to economic circumstances and other factors [21]. Furthermore, failure to achieve the educational arena leads to street children being endorsed with deviant behavior of the society as they fail to pass through the socialization process which in turn plays a pivotal role in shaping character, attitude, and behavior that is utterly in line with the societal norms and values. Thus, it needs cooperative work from governmental and non-governmental organizations to warranty education for those children.

In this study, the most common factors that enforce street children to live in the street were searching for a job, disagreed with parents, peer pressure, and in search of food (poverty). This is in line with studies conducted in Shashemene town, Ethiopia [22], Nepal [3], and Nekemte Town, Ethiopia [23]. One possible explanation is that the socioeconomic status of study settings is similar. According to the World Bank, neither country is a high-income country [24]. As a result, the concerned body anticipated cascading and organizing multi-sector services,

including the creation of job opportunities for addressing the problems of street children, which could benefit and improve the lives of street children [25].

According to the findings of this study, one-third of street children use psychoactive substances. Peer pressure, depression, preventing hunger, preventing fear, coping with cold weather, and growing stronger were the most frequently mentioned reasons by the children. This is consistent with the studies conducted in Ethiopia [22], Nigeria [26], and India [20]. This resemblance could be attributed to the study participants' shared socio-cultural characteristics. This finding implied that there is a clear gap in the implementation of drug and substance policies, which had a significant impact on the health of street children. Even though our country has policies against drug abuse and punishment for those who sell drugs to minors, drugs are readily available and easily accessed by children. This allows street children to purchase and use psychoactive substances [27].

In this study, the majority of the streets got sick in the past month. This finding was in line with a study done in southeastern Ethiopia [17]. This similarity may be attributed to the similarity in the socioeconomic status of the study participants. This study and other evidence outlined that the means of their livelihood were daily laborers and street vendors. This had an impact on their health and this condition of street children necessitates social intervention to address the present health problems of street children by ensuring sustainable livelihood options among them [25].

This study pointed out that more than one-fourth of the participants had a history of sexual intercourse and among those, one-fifth of them had forced sexual intercourse. The history of sexual intercourse may contribute to poor health conditions of street children. Evidence also showed that interventions that aimed to increase the awareness of those populations should be advocated to help this group of children about the dangers and consequences of unprotected sex, STIs, and HIV infection [27].

Respondents under the age of 12 and 13–18 were 3.43 times [AOR = 3.43, 95% CI (1.13,10.42)] and 3.8 times [AOR = 3.80, 95% CI (1.14,10.47)] more likely to become ill in this study when compared to those over the age of 19. As one gets older, one's immune system matures and becomes capable of defending the body against potentially harmful agents. The CD4% level decreased until adolescence, then increased in adults, and finally stabilized in the elderly [28].

In this study, the odds of being sick were 2.12 times higher among participants who had not worked than among those who had worked. These findings are consistent with another study that found strong evidence of an association between unemployment and poorer health outcomes.[29]. Unemployment is almost universally a negative experience, and as a result, it is associated with poor outcomes due to the inability to afford health insurance and insufficient funds to eat a balanced diet. Finally, these hurt one's health.

n this study, the odds of being sick were 2.39 times higher among participants who had a history of sexual intercourse than among those who had no history of sexual intercourse. In fact, most people on the street may engage in unprotected sex, which is dangerous and can lead to a variety of diseases and conditions. In this study, 14.55% of the streets had forced sexual intercourse, and 8.64% had sexual intercourse for financial gain. According to some evidence, unsafe sex is ranked second among the top ten health risk factors in terms of the disease burden it causes. HIV/AIDS, STDs, and unintended pregnancy are just a few of the diseases that can result from unsafe sex [30].

The limitation of the study was the cross-sectional nature of the study (it doesn't show a cause-effect relationship). The other limitation of the study was the small sample size of the study, non-participatory data collection method, and failure to do reunification of the study

participants with their parents. So, we recommend future researchers conduct follow-up studies with larger sample size.

## Conclusion

This study identified the factors that forced the street children to resort to the street way of life including looking for a job and disagreeing with parents. Almost all street children drink alcoholic beverages at different frequencies which exposes them to different health problems. Onefourth of the street children had a history of sexual intercourse including forced sex. Those contributed to the poor health status of street children. Furthermore, factors like age, educational status, presence of work/job, and drug use among the respondents were significantly associated with the health status of street children. To improve the lives of the street children active participation of different stakeholders in drawing a roadmap of health and social interventions is very important.

## Supporting information

**S1 Table. Distribution of sample interviewees by selected sites in the Harar city, eastern Ethiopia, 2021.**
(PDF)

**S1 File. Data collection questionaries.**
(PDF)

**S2 File. The key informant interview guide.**
(PDF)

**S3 File. Street children data used for analysis.**
(DTA)

## Acknowledgments

We would like to extend our deepest thanks to Haramaya University, College of Health and Medical Science staff for providing their constructive support.

## Author Contributions

**Conceptualization:** Degu Abate, Belay Negash, Addisu Alemu, Nebiyu Bahiru, Mohammed Abdurke Kure, Ahmedmenewer Abdu, Amanuel Oljira Dulo, Habtamu Bekele, Saron Bogale, Tewodros Assefa, Barkot Taddesse, Shambel Nigussie, Siraj Aliyi Adem, Gebisa Dirirsa, Yadeta Dessie.

**Data curation:** Degu Abate, Addis Eyeberu, Dechasa Adare, Alemayehu Deressa Wayessa, Ahmedmenewer Abdu, Saron Bogale, Rabuma Belete, Mohammed Muzeyin, Anumein Mohammed, Henock Asfaw, Daniel Alemu, Dawit Yihun, Deribe Bekele.

**Formal analysis:** Addis Eyeberu, Dechasa Adare, Alemayehu Deressa Wayessa, Helina Heluf, Ahmedmenewer Abdu, Kefelegn Bayu, Saron Bogale, Genanaw Atnafe, Anumein Mohammed, Daniel Alemu, Dawit Yihun, Jemal Yusuf Kebira, Saba Hailu, Deribe Bekele.

**Funding acquisition:** Anumein Mohammed.

**Investigation:** Dechasa Adare, Belay Negash, Temam Beshir, Alemayehu Deressa Wayessa, Ahmedmenewer Abdu, Habtamu Bekele, Kefelegn Bayu, Henock Asfaw, Dawit Yihun, Saba Hailu, Galana Mamo.

**Methodology:** Addis Eyeberu, Dechasa Adare, Belay Negash, Ahmedmenewer Abdu, Habtamu Bekele, Saron Bogale, Genanaw Atnafe, Haftu Asmerom, Mesay Arkew, Anumein Mohammed, Henock Asfaw, Shambel Nigussie, Galana Mamo.

**Project administration:** Helina Heluf, Amanuel Oljira Dulo, Kefelegn Bayu, Genanaw Atnafe, Daniel Alemu, Shambel Nigussie, Abduro Godana.

**Resources:** Helina Heluf, Ahmedmenewer Abdu, Kefelegn Bayu, Genanaw Atnafe, Mohammed Muzeyin, Barkot Taddesse, Jemal Yusuf Kebira, Gebisa Dirirsa, Abduro Godana.

**Software:** Belay Negash, Temam Beshir, Adera Debella, Helina Heluf, Amanuel Oljira Dulo, Habtamu Bekele, Genanaw Atnafe, Tewodros Assefa, Mohammed Muzeyin, Anumein Mohammed, Barkot Taddesse, Galana Mamo, Deribe Bekele.

**Supervision:** Addis Eyeberu, Dechasa Adare, Addisu Alemu, Temam Beshir, Adera Debella, Nebiyu Bahiru, Mohammed Abdurke Kure, Tewodros Assefa, Mohammed Muzeyin, Haftu Asmerom, Mesay Arkew, Anumein Mohammed, Henock Asfaw, Barkot Taddesse, Daniel Alemu, Dawit Yihun, Jemal Yusuf Kebira, Saba Hailu, Galana Mamo, Deribe Bekele, Yadeta Dessie.

**Validation:** Addis Eyeberu, Dechasa Adare, Addisu Alemu, Temam Beshir, Adera Debella, Helina Heluf, Mohammed Abdurke Kure, Rabuma Belete, Mohammed Muzeyin, Haftu Asmerom, Mesay Arkew, Anumein Mohammed, Henock Asfaw, Daniel Alemu, Dawit Yihun, Jemal Yusuf Kebira, Siraj Aliyi Adem, Abduro Godana, Deribe Bekele, Yadeta Dessie.

**Visualization:** Addis Eyeberu, Temam Beshir, Adera Debella, Nebiyu Bahiru, Rabuma Belete, Mohammed Muzeyin, Haftu Asmerom, Mesay Arkew, Anumein Mohammed, Henock Asfaw, Siraj Aliyi Adem, Gebisa Dirirsa, Abduro Godana, Yadeta Dessie.

**Writing – original draft:** Addis Eyeberu.

**Writing – review & editing:** Addis Eyeberu, Dechasa Adare, Addisu Alemu, Adera Debella, Nebiyu Bahiru, Mohammed Abdurke Kure, Ahmedmenewer Abdu, Habtamu Bekele, Kefelegn Bayu, Tewodros Assefa, Rabuma Belete, Haftu Asmerom, Barkot Taddesse, Dawit Yihun, Shambel Nigussie, Jemal Yusuf Kebira, Gebisa Dirirsa, Saba Hailu, Yadeta Dessie.

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
