## [Decision Letter · Decision Letter 0]

15 Oct 2021

PONE-D-21-13888Health Conditions of Street Children, Their General Life Conditions, and Reasons Forcing to Street Life in Harar, Eastern Ethiopia. Using Mixed MethodsPLOS ONE

Dear Dr. Eyeberu,

Thank you for submitting your manuscript to PLOS ONE. After careful consideration, we feel that it has merit but does not fully meet PLOS ONE’s publication criteria as it currently stands. Therefore, we invite you to submit a revised version of the manuscript that addresses the points raised during the review process.

Major Revision

We look forward to receiving your revised manuscript.

Kind regards,

Faisal Abbas, PhD

Academic Editor

PLOS ONE

Journal Requirements:

2. Please include additional information regarding the survey or questionnaire used in the study and ensure that you have provided sufficient details that others could replicate the analyses. For instance, if you developed a questionnaire as part of this study and it is not under a copyright more restrictive than CC-BY, please include a copy, in both the original language and English, as Supporting Information. If the original language is written in non-Latin characters, for example Amharic, Chinese, or Korean, please use a file format that ensures these characters are visible."

3. Please state whether you validated the questionnaire prior to testing on study participants. Please provide details regarding the validation group within the methods section."

4. Please include a copy of the interview guide used in the study, in both the original language and English, as Supporting Information, or include a citation if it has been published previously."

5. You indicated that you had ethical approval for your study. In your Methods section, please ensure you have also stated whether you obtained consent from parents or guardians of the minors included in the study or whether the research ethics committee or IRB specifically waived the need for their consent.

7. Your ethics statement should only appear in the Methods section of your manuscript. If your ethics statement is written in any section besides the Methods, please delete it from any other section.

8. Please include a separate caption for each figure in your manuscript

Additional Editor Comments (if provided):

Major Revisions required.

Reviewers' comments:

Reviewer's Responses to Questions

**Comments to the Author**

1. Is the manuscript technically sound, and do the data support the conclusions?

Reviewer #1: No

Reviewer #2: Partly

2. Has the statistical analysis been performed appropriately and rigorously? 

Reviewer #1: No

Reviewer #2: Yes

3. Have the authors made all data underlying the findings in their manuscript fully available?

Reviewer #1: No

Reviewer #2: Yes

4. Is the manuscript presented in an intelligible fashion and written in standard English?

Reviewer #1: No

Reviewer #2: No

5. Review Comments to the Author

Reviewer #1: The research on Health Conditions of Street Children, Their General Life Conditions, and Reasons Forcing to Street Life in Harar, Eastern Ethiopia. Using Mixed Methods, is important in the context of Ethopia. However, I have few suggestions and queries on this manuscript.

Title: Title needs to be rephrased by putting main variables in it.

Abstract: “As it stands now little is known about the extent of the problem including health conditions”. This sentence is unclear, what does it means by “extent of the problem”. In the method you didn’t mention about the type of mixed method used in this study. I don’t see 30 days in February in any year. Please recheck the time duration for data collection. In abstract you are not supposed to write the criteria for the significance level, you can report this in main document. “Overall 220 street children were involved in the study. The most common reason that forced the children to resort to a street way of life is to look a job and quarreled with parents.” These first two lines must be included in the method instead of results. I would suggest you to include both qualitative and quantitative results separately in the result section. In the conclusion, you have reported results, which are redundant to what you have written in result, it is suggested to re write the conclusion of abstract.

Introduction:

The problem was not addressed in a convincing manner. In the introduction section, I couldn’t understand the need to conduct research. It would be better if you explicitly write about the literature, policy, and knowledge gaps in the introduction. In the last paragraph of introduction, authors claim “Now days it is being observed the number of street children is significantly increasing where, a little is known about the extent of the problem including factors leading to being street children and their health status”, without any reference. I also recommend to clearly state the objectives of the study.

Method:

Cross-check the dates of data collection “1-30 February, 2021”. It would be better if you mention the date either in abstract or in methodology, avoid repetition.

Study Design and population: I couldn’t find any description or rationale of choosing mixed method and which mixed method has been chosen. You need to specify research questions identifying quantitative and qualitative method. No information about the sample for qualitative study is reported in this section.

Sample size determination and sampling procedure

What does it means by “Out of this various sit”?

Please provide the reference that the higher number of street children are present in the areas which you have selected for the study. Provide the rationale for applying qualitative method and selection of sample for the qualitative study.

Data collection methods and data collection procedure:

Provide the name of the questionnaires used for the quantitative data collection. If you adapted, then explain the steps/ models used to adapt or contextualize the questionnaire. Also report the reliability and validity of the questionnaires. For qualitative study explain what type of interview guideline was developed and why (e.g., structured, semi-structured, or unstructured). Please share few questions from the interview guideline.

In an article you do not need to provide operational definition however, you can provide the rationale for using specific variables through literature in introduction section.

Data Analysis

What is KII??

Explain triangulation in this study specifically, what is the purpose of triangulation and how would you analysed it?

Statistical Analysis

You have applied multivariate, but you didn’t mention whether the data met assumptions for the multivariate analysis. For qualitative, explain how you develop themes, what procedure or model you followed for finalizing the main themes.

The verbatims stated are not intact. One sentence of the verbatim is not connected with the other sentence. Detail explanation of the verbatim is required. When you clarify the type of mixed method, you will be able to integrate the information from the two studies. Otherwise, the analysis section seems disorganized. Interpretation of the results are also not clear.

In Table 4 “Health status and substance use among study participants in Harar city, Ethiopia”, you have mentioned about depression, do you really think that street children understand what is depression?

Discussion

Instead of reporting percentages in results and discussion, it would be very helpful for the reader to understand if interpretation of the results along the supporting or refuting by previous literature would be reported.

I would suggest that discussion must be organized by giving; 1. argument from literature and culture context, 2. implication of the study and then 3. limitations.

Reviewer #2: • Title is required to be re-phrased.

• Table 1: Under father’s educational status, it is mentioned that around 21% of the children have fathers with education level “highest grade completed”. This category needs to be defined clearly.

• Line 226-232: No need to report frequency since percentages are already given.

• A thorough spellings and grammar check is required for the entire manuscript.

• This paper can be substantially improved by adding more information from the focused group discussions. Qualitative part of the study needs more attention of the authors.

6. PLOS authors have the option to publish the peer review history of their article (what does this mean?). If published, this will include your full peer review and any attached files.

Reviewer #1: **Yes: **Siddrah Irfan

Reviewer #2: No

---

## [Author Response · Author response to Decision Letter 0]

27 Nov 2021

Title: Health Status of Street Children and Reasons for Being Forced to Live on the Streets in Harar, Eastern Ethiopia. Using Mixed Methods

Manuscript ID: PONE-D-21-13888

From: Authors

To: The editor in chief, PLOS ONE

Version: I 

Data: 25/11/2021

Subject: Revision of the manuscript

We appreciate the reviewers' detailed and comprehensive comments. We found the comments to be very helpful, and we appreciate the time and thought that each person put into their constructive comments. We are well aware of the time ,commitment required to provide good reviews and applaud the reviewers for their efforts. We thoroughly revised the paper and responded in detail to the reviewers' questions and comments.The point-by-point description of the changes is provided below.

For editors 

Comment 1: Please ensure that your manuscript meets PLOS ONE's style requirements, including those for file naming. 

Response 1: thank you for your comment. Now the revised version is edited based on PLOS ONE's style requirements.

2. Please include additional information regarding the survey or questionnaire used in the study and ensure that you have provided sufficient details that others could replicate the analyses. For instance, if you developed a questionnaire as part of this study and it is not under copyright more restrictive than CC-BY, please include a copy, in both the original language and English, as Supporting Information. If the original language is written in non-Latin characters, for example, Amharic, Chinese, or Korean, please use a file format that ensures these characters are visible."

Response 2: Thank you very much. We have now added more information about the survey.

3. Please state whether you validated the questionnaire before testing on study participants. Please provide details regarding the validation group within the methods section."

Response 3: Thank you for your input. It has been corrected and is now part of the revised document.

4. Please include a copy of the interview guide used in the study, in both the original language and English, as Supporting Information, or include a citation if it has been published previously."

Response 4: Thank you. We now include it as an addendum.

5. You indicated that you had ethical approval for your study. In your Methods section, please ensure you have also stated whether you obtained consent from parents or guardians of the minors included in the study or whether the research ethics committee or IRB specifically waived the need for their consent.

Response 5: Thank you for your input. It has been corrected and is now part of the revised document. Consent was obtained from the minors' parents or guardians..

6. In your Data Availability statement, you have not specified where the minimal data set underlying the results described in your manuscript can be found. PLOS defines a study's minimal data set as the underlying data used to reach the conclusions drawn in the manuscript and any additional data required to replicate the reported study findings in their entirety. All PLOS journals require that the minimal data set be made fully available.

Response 6: Thank you for your comment. It is corrected and incorporated into the revised document. 

7. Your ethics statement should only appear in the Methods section of your manuscript. If your ethics statement is written in any section besides the Methods, please delete it from any other section.

Response 7: Thank you for your comment. It is corrected and incorporated into the revised document. Ethics statement was omitted from declaration section.

8. Please include a separate caption for each figure in your manuscript

Response 8: We are intrigued by your comment, and we accept it; it has been corrected and incorporated into the revised document.

For reviewer 1 

Comment 1. The research on Health Conditions of Street Children, Their General Life Conditions, and Reasons Forcing to Street Life in Harar, Eastern Ethiopia. Using Mixed Methods is important in the context of Ethiopia. However, I have a few suggestions and queries on this manuscript

Response 1: thank you for your comment. The significance of this study in the context of Ethiopia is obvious for a variety of reasons, which are briefly discussed below. I believe it is critical because numerous factors lead a person to leave his or her home and join the life of a street person. It is critical because not everyone in Ethiopia lives on the street; if that were the case, we would not need to study because everyone was affected by these serious issues. So, to mitigate the problems and devise solutions that will undoubtedly address a long-standing issue, this research is crucial.

Comment 2. Title: The title needs to be rephrased by putting main variables in it.

 Response 2: thank you for your nice comment. It is corrected as the Health Status of Street Children and Reasons for Being Forced to Live on the Streets in Harar, Eastern Ethiopia. Using Mixed Methods. 

Comment 3. “As it stands now little is known about the extent of the problem including health conditions”. This sentence is unclear, what does it mean by “extent of the problem”.

 Response 3: thank you for your comment and questions. When we talk about the extent of the problem, we're referring to the prevalence of Streetism, which includes health conditions. This sentence can be paraphrased as “Currently, little is known about the prevalence of the Streetism, including health conditions.

Comment 4. In the method, you didn’t mention the type of mixed method used in this study.

 Response 4: We are intrigued by your comment, and we accept it; it has been corrected and incorporated into the revised document. The study designs were quantitative cross-sectional studies and qualitative phenomenological studies.

Comment 5. I don’t see 30 days in February in any year. Please recheck the time duration for data collection

Response 5: Thank you for your comment and question. We apologize for such kind of error. Now it is corrected and incorporated in the revised manuscript. 

Comment 6. In the abstract you are not supposed to write the criteria for the significance level, you can report this in the main document

 Response 6: Thank you for your comment and suggestion. It is corrected and incorporated in the revised manuscript. 

Comment 7. Overall, 220 street children were involved in the study. The most common reason that forced the children to resort to a street way of life is to look for a job and quarreled with parents.” These first two lines must be included in the method instead of results.

Response 7: Thank you for your insightful and wonderful comment. We accept your comment and we incorporate it into the revised manuscript. 

Comment 8. I would suggest you include both qualitative and quantitative results separately in the result section.

Response 8: Thank you for your comment. We accept it and corrected it in the revised manuscript. 

Comment 9. In the conclusion, you have reported results, which are redundant to what you have written in the result, it is suggested to rewrite the conclusion of the abstract.

Response 9: Thank you for your specific comment. We accept it and corrected it in the revised manuscript.

Comment 10. The problem was not addressed convincingly. In the introduction section, I couldn’t understand the need to conduct research. It would be better if you explicitly write about the literature, policy, and knowledge gaps in the introduction.

Response 10. Thank you for your insightful comment. We revised the introduction section based on your recommendations. 

Comment 11. In the last paragraph of the introduction, the authors claim “Nowadays it is being observed the number of street children is significantly increasing were, a little is known about the extent of the problem including factors leading to being street children and their health status”, without any reference.

Response 11. Thank you for your specific comment. We revised the citation. Now it is cited and a change was made in the revised manuscript. 

Comment 12. I also recommend stating the objectives of the study.

 Response 12. Thank you for your wonderful and eye-catching comments. We accept it and revise accordingly. 

Comment 13. Cross-check the dates of data collection “1-30 February 2021”.

 Response 13: Thanks for your eagle eyes we are already changed it 

Comment 14. It would be better if you mention the date either in the abstract or in methodology, avoiding repetition.

Response 14: thank you for your insightful suggestion. Now the revised version is edited and corrected.

 Comment 15: I couldn’t find any description or rationale for choosing a mixed-method and which mixed method has been chosen. 

Response 15: thank you for your comment. Bringing together qualitative and quantitative research and data in a single study. Mixed-method research is a hybrid that emerged from the most appropriate combination of quantitative and qualitative research methods for a research project. It aimed to maximize the strengths and minimize the weaknesses of the two research methods, increasing the validity of research findings and providing other benefits. As a result, we attempted to use qualitative and quantitative research methods to improve the validity and credibility of the results in particular, as well as the overall research in general. To elaborate further, we employ a qualitative study design based on the Phenomenological approach that aims to develop a complete, accurate, clear, and articulate description and understanding of a particular human experience or experiential moment (Schacht, 1972).

Comment 16: You need to specify research questions identifying quantitative and qualitative methods.

Response 16: thank you for your comment. It is corrected and incorporated in the revised manuscript.

Comment 17: No information about the sample for the qualitative study is reported in this section.

 Response 17: Thank you for your comment. The sample size was determined based on the level of information saturation and the variety of ideas among the sub-groups. Data saturation involves sampling until no new information is obtained and redundancy is achieved. We conducted twelve focus group discussions (FGDs) with six participants from various groups and 6 key informants.

Comment 18: What does it mean by “Out of this various sit”?

Response 18: Thank you for your question and comment. This means that of the seven areas mentioned... we attempted to paraphrase the sentences containing this phrase as follows. Street children can be found in a variety of locations throughout Harar. Seven are chosen on purpose because there is a higher concentration of street children in these areas of town known as Arategna, Ajip, Canal, Botte, Shewa ber, Feres Megala, and Andegan Menged. This was accomplished through observation and consultation with key informants working with street children.

Comment 19: Please provide the reference that the higher number of street children are present in the areas which you have selected for the study. 

Response 19: Thank you for your comment. This was the data source obtained from the social affairs office 

Table 1: Distribution of sample interviewees by selected sites in the Harar city

S. No Name of the Sub town Total of the registered population Number of the interviewees

1 Arategna 50 23

2 Agip 61 28

3 Canal 53 24

4 Bote 92 42

5 Shewa Ber 101 46

6 Andegan Menged 99 44

7 Feres megala 46 21

 Total 502 228

Comment 20: Provide the rationale for applying the qualitative method and selection of sample for the qualitative study.

Response 20: Thank you for your comment and suggestion. There are plenty of rationale for applying the qualitative method. To mention a few of them, to obtain more in-depth information on cases (subjective), to validate and confirm the information gathered, and to increase the validity of research findings. It is useful in establishing meaningful relationships between the researcher and ordinary people, which allows for a better understanding of their world and interpretation of the results in terms of a qualitative approach. So, an attempt was made to apply qualitative methods to study these vulnerable groups. Regarding the selection of the sample, participants were selected based on criteria based purposive sampling to get important information from lived experience. 

Comment 21: Provide the name of the questionnaires used for the quantitative data collection. If you adapted, then explain the steps/ models used to adapt or contextualize the questionnaire. 

Response 21: Dear Reviewer, when adapting the questioners, we kept three things in mind: Language, Pretesting, and the specific question that will be modified

Steps one: We translated it while keeping the purpose of the questionnaire and the intent of the questions in mind. It was done by group members who speak both languages fluently translate it. To ensure the translation's accuracy, the questionnaire was translated back into English by someone who had not seen the original version and was unfamiliar with the questionnaire's context. The back-translated version is then compared to the original, and any meaning differences are corrected.

 Step two: To ensure a cross-validity we tried to interview a set of respondents in English and another set in the local language such as Afaan Oromo and Amharic, their answers were then compared to detect differences in understanding.

Step three: Pretesting: The goals of pre-testing are well known: to identify questions that are poorly understood, ambiguous, or elicit hostile or other undesirable responses. We attempted to conduct a pretest using the already-translated questionnaire. We tried to implement all the steps in pretesting such as obtaining an evaluation of a questionnaire and testing the revised questionnaire through its paces on friends, colleagues, and so on.

Comment 22: Also report the reliability and validity of the questionnaires. 

Response 22: When choosing a survey instrument, reliability and validity must be taken into account. The consistency with which an instrument produces the same results across multiple trials are referred to as its reliability. The degree to which an instrument measures what it was designed to measure is known as its validity. Statistically, we performed Cronbach's Alpha, which is a measure used to assess the quality of our employed instruments. The result was 0.87, which was within acceptable ranges.

Comment 23: For qualitative study explain what type of interview guideline was developed and why (e.g., structured, semi-structured, or unstructured). Please share a few questions from the interview guideline.

Response 23: Thank you for your comment. In our study, a semi-structured interview guideline was used. Dear reviewers, in response to your humble request, please find below a few questions from the interview guideline. 

What relationship does your organization/office have with the children of the street?

To what extent is the government concerned about the situation of the street children?

What are the main problems that they face?

What services that your office currently offering to the street children?

Comment 24: In an article, you do not need to provide an operational definition however, you can provide the rationale for using specific variables through literature in the introduction section.

Response 24: Dear reviewer, we gratefully received your constructive suggestion and omitted it (operational definition) from an article.

Comment 25: What is KII??

Response 25: Dear reviewers, please accept our apologies for such a clumsy error when we say KII we mean key informant interview!!

Comment 26: Explain triangulation in this study specifically, what is the purpose of triangulation and how would you have analyzed it?

Response 26: Dear Respected Reviewer, as we all know, triangulation is a well-known technique for increasing the credibility and validity of research findings. Triangulation is a research study can help to ensure that fundamental biases caused by the use of a single method or observer are overcome by combining theories, methods, or observers. In our study, we attempted to use this robust method to improve the trustworthiness and accuracy of our findings. As a result, methodological (both qualitative and quantitative methods) and data triangulation (from different sources using different data collection methods) were carried out to improve the credibility and validity of the research findings

Comment 27: You have applied multivariate, but you didn’t mention whether the data met assumptions for the multivariate analysis.

Response 27: Thank you for your constructive comment. Dear reviewer, before performing a multivariate regression analysis, we verified that all assumptions and data met the criteria, such as

• The relationship between independent variables and dependent variables was linear (which was checked by producing the scatter plots)

• There was no multicollinearity in our data (on collinearity diagnosis we tried to see VIF standard error and Tolerance score)

• Values of the residuals were found normality distributed (Tested by looking at P-P plots from the model)

• A variance of the residuals was found constant (which was checked by producing the scatter plots)

• No influential cases were biasing our models (checked by looking at Cooks distance)

Comment 28: For qualitative, explain how you develop themes, what procedure or model you followed for finalizing the main themes.

Response 28: thank you for your comment. We follow the inductive thematic analysis procedures. First deeply read and read again and again. Then we code the ideas. Then we categorize similar ideas and then it was developed into their thematic areas. To do this we carefully examine the data to identify common themes – topics, ideas, and meaning patterns that appear repeatedly. To finalize the main themes, we followed the procedures such as familiarization, coding, generating themes, reviewing themes, defining and naming them, and writing up.

Comment 29: The verbatims stated are not intact. One sentence of the verbatim is not connected with the other sentence. Detail explanation of the verbatim is required. 

Response 29: Thank you for your comment and suggestion. It is corrected and incorporated in the revised manuscript. 

Comment 30: When you clarify the type of mixed-method, you will be able to integrate the information from the two studies. Otherwise, the analysis section seems disorganized. Interpretation of the results is also not clear.

Response 30: Thank you for your comment. It is corrected and incorporated into the revised manuscript. 

Comment 31: In Table 4 “Health status and substance use among study participants in Harar city, Ethiopia”, you have mentioned depression, do you think that street children understand what is depression?

Response 31: Dear Reviewer, yes, they have no idea what depression entails. We asked them in their native language about depression; they had no idea about the scientific terms or nomenclatures of depression that here we used to report to the scientific world.

Comment 32: Instead of reporting percentages in results and discussion, it would be very helpful for the reader to understand if the interpretation of the results along the supporting or refuting by previous literature would be reported

Response 32: Thank you for your comment. Now it is modified based on the comment provided. 

Comment 33: I would suggest that discussion must be organized by giving; 1. argument from literature and culture context, 2. implication of the study, and then 3. limitations.

Response 33: Thank you for your comment. It is corrected and incorporated in the revised manuscript. 

For reviewer 2

Comment 1. The title is required to be re-phrased

 Response 1. Thank you for your specific comment. It is corrected and incorporated into the revised manuscript. 

Comment 2. Table 1: Under father’s educational status, it is mentioned that around 21% of the children have fathers with education level “highest grade completed”. This category needs to be defined clearly.

 Response 2. thank you for your specific comment. We accepted the comment and it is corrected in the revised document. According to our study, this is to mean that from grade secondary level (8-10) to preparatory level (10-12)

Comment 3. Line 226-232: No need to report frequency since percentages are already given. 

 Response 3. Thank you for your nice comment. We incorporate comments in the revised manuscript. 

Comment 4. A thorough spelling and grammar check is required for the entire manuscript.

 Response 4. Thank you for this comment. It is corrected and incorporated in the revised manuscript.

Comment 5. This paper can be substantially improved by adding more information from the focused group discussions. The qualitative part of the study needs more attention from the authors.

Response 5. Thank you for your insight full comment. It is corrected on the revised document.

Thank you

---

## [Decision Letter · Decision Letter 1]

20 Dec 2021

PONE-D-21-13888R1Health status of street children and reasons for being forced to live on the streets in Harar, Eastern Ethiopia. Using mixed methodsPLOS ONE

Dear Dr. Eyeberu,

Thank you for submitting your manuscript to PLOS ONE. After careful consideration, we feel that it has merit but does not fully meet PLOS ONE’s publication criteria as it currently stands. Therefore, we invite you to submit a revised version of the manuscript that addresses the points raised during the review process.

Major Revisions

We look forward to receiving your revised manuscript.

Kind regards,

Faisal Abbas, PhD

Academic Editor

PLOS ONE

Additional Editor Comments:

Major Revisions.

Reviewers' comments:

Reviewer's Responses to Questions

**Comments to the Author**

1. If the authors have adequately addressed your comments raised in a previous round of review and you feel that this manuscript is now acceptable for publication, you may indicate that here to bypass the “Comments to the Author” section, enter your conflict of interest statement in the “Confidential to Editor” section, and submit your "Accept" recommendation.

Reviewer #1: (No Response)

Reviewer #2: All comments have been addressed

2. Is the manuscript technically sound, and do the data support the conclusions?

Reviewer #1: No

Reviewer #2: Yes

3. Has the statistical analysis been performed appropriately and rigorously? 

Reviewer #1: No

Reviewer #2: Yes

4. Have the authors made all data underlying the findings in their manuscript fully available?

Reviewer #1: Yes

Reviewer #2: Yes

5. Is the manuscript presented in an intelligible fashion and written in standard English?

Reviewer #1: No

Reviewer #2: Yes

6. Review Comments to the Author

Reviewer #1: Health status of street children and reasons for being forced to live on the streets in Harar, Eastern Ethiopia. Using mixed methods

This research is important in the context of Ethopia and much needed in current scenario. What and how this research adds in the literature and help policy maker to work for the betterment of street children. I can see improvement in the manuscript, but I have few comments and suggestion for your manuscript, which are as follows:

Abstract

1. Introduction is well written

2. In Method section, you need to report type of mixed method (e.g., sequential or parallel). I have few questions about phenomenological design (See method section).I want to know the reason for conducting interview for quantitative research design and for qualitative you have done focus group. There is no need to mention softwares in abstract.

3. Results: You didn’t mention result heading. For abstract you can explain results by integrating findings of quantitative and qualitative results.

4. Conclusion is good.

Main Manuscript

Introduction:

You have explained the problem in the introduction, but you can add literature from the similar context or from the courtiers which have similar culture and context, to explain causes that force street children to stay on streets.

Material and Method

1. Study setting, and period is explained quite well

2. Explain type of mixed method design. In case of phenomenological study, you need to explain the lived experience. What experience you are studying and how.

3. Sample size determination is fine for quantitative study but for qualitative specially for phenomenological studies the sample size needs to be selected carefully.

4. You didn’t mention name of questionnaires or scales for the quantitative study. Which instrument you adapted. You also have to mention questions of the interview guideline for the focus group.

Results:

This section is quite improved; however, I couldn’t understand the integration point for both qualitative and quantitative studies. Specially, if you are reporting that you did phenomenological study, you need to report lived experiences of the street children.

Discussion

This study has rich data on health status of street children that need to be explained and interpreted in detail. There is multiple limitation for the study that you need to report. You need to explain the future recommendation as well.

Reviewer #2: (No Response)

7. PLOS authors have the option to publish the peer review history of their article (what does this mean?). If published, this will include your full peer review and any attached files.

Reviewer #1: **Yes: **Siddrah Irfan

Reviewer #2: No

---

## [Author Response · Author response to Decision Letter 1]

2 Feb 2022

Authors’ Response to Reviewers’ comments and Suggestions

Manuscript ID: PONE-D-21-13888

Journal: PLOS ONE

From: Authors

To: The editor in chief, PLOS ONE

Version: II 

Data: 21/12/2021

Dear Reviewers

Thank you so much for giving us an opportunity to submit a revised draft of our manuscript entitled “Health Status of Street Children and Reasons for Being Forced to Live on the Streets in Harar, Eastern Ethiopia. Using Mixed Methods to this highly visible impact factor and peer-reviewed Journal. We appreciate the time and effort that you and the reviewers dedicated to providing feedback on our manuscript. We are very grateful for the insightful comments and valuable improvements to our premature paper. We have incorporated most of the suggestions and comments made by the handling editor and reviewers. All comments and suggestions are clearly stated and well addressed (a point-by-point to the reviewer’s comments and concerns). These changes are highlighted in Bright Green color within the clean revised manuscript. 

Authors’ Response to Reviewer 1’s Comments and Suggestions

Title: Health Status of Street Children and Reasons for Being Forced to Live on the Streets in Harar, Eastern Ethiopia. Using Mixed Methods 

To: Reviewer 1

From: Addis Eyeberu (Corresponding Author)

Subject: Submission of Incorporated Comments and Suggestions

First and foremost, we would like to thank you for your constructive and valuable comments and helpful suggestions that helped us to improve and enrich our premature manuscript. Here is the table we have pointed out how authors incorporated your valuable comments, suggestions, and concerns one by one. 

Reviewer’s Comments to the Authors Authors’ Responses to Reviewer’s comments

Reviewer’s General Comments Overall, thank you very much for your positive and constructive suggestions

1. If the authors have adequately addressed your comments raised in a previous round of review and you feel that this manuscript is now acceptable for publication, you may indicate that here to bypass the “Comments to the Author” section, enter your conflict-of-interest statement in the “Confidential to Editor” section, and submit your "Accept" recommendation.

Reviewer #1: (No Response) Thank you very much for your valuable inputs in advancing this manuscript. Now we tried to address all comments and it is incorporated in the revised manuscript. 

2. Is the manuscript technically sound, and do the data support the conclusions?

Reviewer #1: No Thank you very much!! Now we have tried our best to utilize the comments and suggestions to make the manuscript technically sound and the data support the conclusion. 

3. Has the statistical analysis been performed appropriately and rigorously?

Reviewer #1: No Thank you so much. We would like to thank you for your inputs and constructive suggestion. Now, we enriched and revised the paper after initial submission.

4. Have the authors made all data underlying the findings in their manuscript fully available?

Reviewer #1: Yes Thanks! 

5. Is the manuscript presented in an intelligible fashion and written in standard English?

Reviewer #1: No Thank you very much for such valuable intellectual input. The reviewer is correct. The authors critically considered this input. Now, we thoroughly revised and edited the whole parts of our manuscript and extensively corrected all copy-editing errors in the clean revised manuscript. The authors also sent the manuscript to a language expert/editor who critically reviewed, edited, and corrected all language-related errors made in a submitted manuscript.

6. Specific Review’s Comments to the Authors Authors Responses to Reviewers’ Specific comments

# Reviewer 1. What and how this research adds to the literature and help policymaker work for the betterment of street children. Thank you for your insightful questions about how this research adds to existing literature and what it means for policymakers. This, I believe, is a critical question that must be addressed because it is both the core and the yolk of research's ultimate finding. 

This study's findings have a significant and profound impact on the existing literature in one way or another. Here are just a few instances from a long list.

o Despite being a pocket study, this study attempted to point out the health status of street children. Because they are vulnerable groups or segments of our population, they have received some attention. As a result, we humbly believe that we are in a position to add this small droplet to the vast ocean of previously acquired knowledge.

o We also endeavored to demonstrate why street children are forced to live on the streets. Only studying the health status of our target population is insufficient; as we found them on the street, we need to find out what factors compelled them to leave their home and live on the street.

o This study also demonstrates how common divorce was among the parents of street children, which was a previously understudied area. As a result, we attempted to see this and demonstrate how prevalent the problem is.

This research finding could play a critical role in improving the lives of street children for policymakers in some way. to name a few.

o Policymakers must take steps toward reuniting street children with their families. For any child longing to be reunited with his or her family, the emotional and psychological hurdles can be enormous. This is especially true for children who have been reintegrated into their families and communities after spending time on the streets as a result of a variety of issues such as family breakdown, parent death, and poverty. As a result, this study throws at least one stone in the direction of changing the current situation. 

o By identifying the causes of streetism, policymakers can intervene on the factors that push children to become street children. Poverty, a lack of education, abuse, and a lack of parental care all contribute to the plight of street children. Push-factors such as abuse, domestic violence, or dysfunctional family relationships are common among street children. It may even come to the point where their situation at home becomes unbearable and they choose to live on the street. Therefore, as preventative work reduces the number of children living on the street, organizations must make significant efforts early on.

o To curb the phenomenon of street children, policymakers should take steps to socialize this vulnerable segment of our society. This study plays an important role in this regard.

o This study assists policymakers in working on the education of street children, allowing those vulnerable segments of the population to shape character, attitude, and behavior that is completely consistent with societal norms and values.

o It enables policymakers to devise alternative solutions for street children on how to live their lives in ways other than engaging in criminal activity.

o This study appears to assist policymakers in taking action on substance use among street children, which leads to widespread problems for them.

2. You need to report the type of mixed method (e.g., sequential or parallel). Thank you very much for your countless effort to review our premature manuscript. Now we revised the manuscript based on the comment and suggestions. The comment is incorporated in the revised manuscript. 

3.“I want to know the reason for conducting an interview for quantitative research design and for qualitative you have done focus group. Dear, Reviewer, thank you very much for your countless effort. Dear respected reviewer, to be clear from the very outset, we used interviewer-administered questionnaires to collect quantitative data about the participants, while focus group discussions and key informants were used to collect qualitative data. And we have a plenty of reasons for conducting interviewer-administered questionnaires for quantitative research designs, and focus groups for qualitative research designs. We attempted to see and elaborate separately in the sections that follow.

Reason why we undertook Focus group discussion for qualitative research design 

o In order to gather information in a short period of time 

o To uncover participants' perceptions and values about the problem at hand (Health status of street children and reasons forced them to live on the street)

o So as to dig out and obtain multiple perspectives on the same topic

o To collect data on collective opinions. 

Collecting information from carefully chosen groups of people 

4. There is no need to mention software in the abstract. Thank you so much for your valuable and insightful comments. It is corrected and incorporated in the revised manuscript. 

5. You didn’t mention the result heading. For abstract, you can explain results by integrating findings of quantitative and qualitative results. Thank you for your comment and question. We apologize for such kinds of errors. Now it is corrected and incorporated in the revised manuscript. 

6. You have explained the problem in the introduction, but you can add literature from a similar context or from the courtiers which have similar culture and context, to explain causes that force street children to stay on the streets. Thank you for your comment and suggestion. It is corrected and incorporated in the revised manuscript. Now the introduction is revised based on the comments and suggestions. 

7. Explain the type of mixed-method design. In the case of phenomenological study, you need to explain the lived experience. What experience you are studying and how.

 Thank you for your insightful and wonderful comment. We accept your comment and we incorporate it into the revised manuscript. Since we conduct FGD, their lived experiences were studied and we incorporated the results in the revised manuscript.

8. Sample size determination is fine for quantitative study but for qualitative especially for phenomenological studies, the sample size needs to be selected carefully.

 Thank you for your comment. We accept it and corrected it in the revised manuscript. 

9. You didn’t mention the name of questionnaires or scales for the quantitative study. Which instrument you adapted. You also have to mention questions of the interview guideline for the focus group.

 Thank you for your specific comment. Since we adapted and modified the questionaries from different kinds of literature into the local context, it doesn’t have a specific name however it is a semi-structured questionary. For FGD the sample questions include

Focus group discussion items

1. What do you think are the major causes of child Streetism? 

2. Can you give details of the difficulties that street children face?

3. can you give details of coping mechanisms of street livelihood?

4. What do you think the multiple agencies should do to help street children? 

5. Do you believe the intervention of the agencies dealing with street children and their families is adequate? if not what do you recommend as an improvement 

10. Result: This section is quite improved; however, I couldn’t understand the integration point for both qualitative and quantitative studies. Especially, if you are reporting that you did a phenomenological study, you need to report lived experiences of the street children.

 Thank you for your insightful comment. We revised the result section based on your recommendations.

11. Discussion: This study has rich data on the health status of street children that need to be explained and interpreted in detail. There are multiple limitations to the study that you need to report. You need to explain the future recommendation as well.

 Thank you for your specific comment. We revised the discussion. Now it is corrected and incorporated in the revised manuscript.

End of authors responses for Reviewer 1

Authors’ Response to Reviewer 2’s Comments and Suggestions

Reviewer’s General Comments Overall, thank you very much for your positive and constructive suggestions

1. If the authors have adequately addressed your comments raised in a previous round of review and you feel that this manuscript is now acceptable for publication, you may indicate that here to bypass the “Comments to the Author” section, enter your conflict-of-interest statement in the “Confidential to Editor” section, and submit your "Accept" recommendation.

Reviewer #2: All comments have been addressed

 Thank you very much for your valuable inputs in advancing this manuscript. 

2. Is the manuscript technically sound, and do the data support the conclusions?

Reviewer #2: Yes Thank you very much!!

3. Has the statistical analysis been performed appropriately and rigorously?

Reviewer #2: Yes Thank you so much. We would like to thank you for your appreciation and constructive suggestion. 

4. Have the authors made all data underlying the findings in their manuscript fully available?

Reviewer #2: Yes Thanks! 

5. Is the manuscript presented in an intelligible fashion and written in standard English?

Reviewer #2: Yes Thank you very much for such valuable intellectual input. 

End of authors responses for Reviewer 2

---

## [Decision Letter · Decision Letter 2]

23 Feb 2022

PONE-D-21-13888R2Health status of street children and reasons for being forced to live on the streets in Harar, Eastern Ethiopia. Using mixed methodsPLOS ONE

Dear Dr. Eyeberu,

Thank you for submitting your manuscript to PLOS ONE. After careful consideration, we feel that it has merit but does not fully meet PLOS ONE’s publication criteria as it currently stands. Therefore, we invite you to submit a revised version of the manuscript that addresses the points raised during the review process.

Minor Revision Please submit your revised manuscript by Apr 09 2022 11:59PM. If you will need more time than this to complete your revisions, please reply to this message or contact the journal office at plosone@plos.org. Please include the following items when submitting your revised manuscript:A rebuttal letter that responds to each point raised by the academic editor and reviewer(s). You should upload this letter as a separate file labeled 'Response to Reviewers'.A marked-up copy of your manuscript that highlights changes made to the original version. You should upload this as a separate file labeled 'Revised Manuscript with Track Changes'.An unmarked version of your revised paper without tracked changes. You should upload this as a separate file labeled 'Manuscript'.If applicable, we recommend that you deposit your laboratory protocols in protocols.io to enhance the reproducibility of your results. Protocols.io assigns your protocol its own identifier (DOI) so that it can be cited independently in the future. For instructions see: https://journals.plos.org/plosone/s/submission-guidelines#loc-laboratory-protocols. Additionally, PLOS ONE offers an option for publishing peer-reviewed Lab Protocol articles, which describe protocols hosted on protocols.io. Read more information on sharing protocols at https://plos.org/protocols?utm_medium=editorial-email&utm_source=authorletters&utm_campaign=protocols.

We look forward to receiving your revised manuscript.

Kind regards,

Faisal Abbas, PhD

Academic Editor

PLOS ONE

Journal Requirements:

Additional Editor Comments (if provided):

Minor Revisions.

Reviewers' comments:

Reviewer's Responses to Questions

**Comments to the Author**

1. If the authors have adequately addressed your comments raised in a previous round of review and you feel that this manuscript is now acceptable for publication, you may indicate that here to bypass the “Comments to the Author” section, enter your conflict of interest statement in the “Confidential to Editor” section, and submit your "Accept" recommendation.

Reviewer #1: All comments have been addressed

Reviewer #2: All comments have been addressed

2. Is the manuscript technically sound, and do the data support the conclusions?

Reviewer #1: Partly

Reviewer #2: Yes

3. Has the statistical analysis been performed appropriately and rigorously? 

Reviewer #1: (No Response)

Reviewer #2: Yes

4. Have the authors made all data underlying the findings in their manuscript fully available?

Reviewer #1: (No Response)

Reviewer #2: Yes

5. Is the manuscript presented in an intelligible fashion and written in standard English?

Reviewer #1: Yes

Reviewer #2: Yes

6. Review Comments to the Author

Reviewer #1: Authors worked very hard to improve the manuscript, which is depicted in the manuscript. However, I have few comments which authors may incorporate.

1. Please double check language and typing mistakes in the article.

2. Mention the name of the questionnaire you adapted and translated (a (Afan, Oromo, and Amharic)

3. What is the validity and reliability of original version of a (Afan, Oromo, and Amharic)

4. Cross check whether in sequential mixed method, you will have to collect data from the same respondents or not and share the reference of mixed method you have followed.

Reviewer #2: (No Response)

7. PLOS authors have the option to publish the peer review history of their article (what does this mean?). If published, this will include your full peer review and any attached files.

Reviewer #1: **Yes: **Siddrah Irfan

Reviewer #2: No

---

## [Author Response · Author response to Decision Letter 2]

28 Feb 2022

Authors’ Response to Reviewers’ comments and Suggestions

Manuscript ID: PONE-D-21-13888

Journal: PLOS ONE

From: Authors

To: The editor in chief, PLOS ONE

Version: III 

Date: 26/2/2022

Dear Reviewers

Thank you so much for allowing us to submit a third revised draft of our manuscript entitled “Health Status of Street Children and Reasons for Being Forced to Live on the Streets in Harar, Eastern Ethiopia. Using Mixed Methods to this highly visible impact factor and peer-reviewed Journal. We appreciate the time and effort that you and the reviewers dedicated to providing feedback on our manuscript. We are very grateful for the insightful comments and valuable improvements to our premature paper. We have incorporated most of the suggestions and comments made by the handling editor and reviewers. All comments and suggestions are clearly stated and well addressed (a point-by-point to the reviewer’s comments and concerns). These changes are highlighted in Bright Green color within the clean revised manuscript. 

Authors’ Response to Editors and Reviewer 1’s Comments and Suggestions

Title: Health Status of Street Children and Reasons for Being Forced to Live on the Streets in Harar, Eastern Ethiopia. Using Mixed Methods 

To: Editor and Reviewer 1

From: Addis Eyeberu (Corresponding Author)

Subject: Submission of Incorporated Comments and Suggestions

First and foremost, we would like to thank you for your constructive and valuable comments and helpful suggestions that helped us to improve and enrich our premature manuscript. Here is the table we have pointed out how authors incorporated your valuable comments, suggestions, and concerns one by one. 

Editors comment to Authors Authors’ Responses to Editor’s comments

Please review your reference list to ensure that it is complete and correct. If you have cited papers that have been retracted, please include the rationale for doing so in the manuscript text, or remove these references and replace them with relevant current references. Any changes to the reference list should be mentioned in the rebuttal letter that accompanies your revised manuscript. If you need to cite a retracted article, indicate the article’s retracted status in the References list and also include a citation and full reference for the retraction notice. Thank you for your comment. Now the authors reviewed the references list. The reference lists were revised to make them complete and correct based on your recommendation and journal requirement. No article has been retracted. 

End of editor’s comment 

Reviewer’s Comments to the Authors Authors’ Responses to Reviewer’s comments

Reviewer’s General Comments Overall, thank you very much for your positive and constructive suggestions

1. If the authors have adequately addressed your comments raised in a previous round of review and you feel that this manuscript is now acceptable for publication, you may indicate that here to bypass the “Comments to the Author” section, enter your conflict-of-interest statement in the “Confidential to Editor” section, and submit your "Accept" recommendation.

Reviewer #1: All comments have been addressed Thank you very much again for your valuable and wonderful input in advancing this manuscript. 

2. Is the manuscript technically sound, and do the data support the conclusions?

Reviewer #1: Partly Thank you very much!! Now we have tried our best to utilize the comments and suggestions to make the manuscript technically sound and the data support the conclusion. 

3. Has the statistical analysis been performed appropriately and rigorously?

Reviewer #1: No Response Thank you so much. We would like to thank you for your input and constructive suggestion. Now, we enriched and revised the paper based on your constructive input.

4. Have the authors made all data underlying the findings in their manuscript fully available?

Reviewer #1: No Response Thanks! we provide all necessary data to the journal and we made all data underlying the findings in their manuscript fully available. 

5. Is the manuscript presented in an intelligible fashion and written in standard English?

Reviewer #1: Yes Thank you very much for such valuable intellectual input. 

6. Specific Review’s Comments to the Authors Authors' Responses to Reviewers’ Specific comments

# Reviewer 1. Please double-check the language and typing mistakes in the article. Thank you for your insightful comments. Now the authors checked the language and typing errors. The authors critically considered this input. Now, we thoroughly revised and edited the whole parts of our manuscript and extensively corrected all copy-editing errors in the clean revised manuscript. The authors also sent the manuscript to a language expert/editor who critically reviewed, edited, and corrected all language-related errors made in a submitted manuscript.

2. Mention the name of the questionnaire you adapted and translated (Afan Oromo, and Amharic) Thank you very much for your countless effort to review our manuscript. Now we revised the manuscript based on the comments and suggestions. The comments are incorporated in the revised manuscript. We tried to do our best to adapt a tool named “the health status of street children survey” from measure evaluation which was funded by USAID and undertaken with the title of “children in Adverse Situations Indicators and Survey Tools”, available at https://www.measureevaluation.org/resources/publications/tl-19-35e.html and from previous studies (BSS tool), which was done by Demelash Habtamu, Addisie Adamu, titled with "Assessment of Sexual and Reproductive Health Status of Street Children in Addis Ababa", Journal of Sexually Transmitted Diseases, vol. 2013, Article ID 524076, 20 pages, 2013. https://doi.org/10.1155/2013/524076

3. What is the validity and reliability of the original version of (Afan Oromo, and Amharic) Dear reviewer, thank you very much for your countless effort. Dear respected reviewer, we tried to mention the validity and reliability of the original tools (English version) in our first version of comments both on revised documents and response to reviewers. In that time, we considered the whole measure we are expected to do before administering data collection tool to the participants so as to enhance and improve the quality of the study. It is clearly known that, the consistency with which an instrument produces the same results across multiple trials are referred to as its reliability. The degree to which an instrument measures what it was designed to measure is known as its validity. Regarding to the English version of the tool we tried to check the validity and reliability statistically. To this end, we performed Cronbach's Alpha, which is a measure used to assess the quality of our employed instruments. The result was 0.87, which was within acceptable ranges. Nevertheless, we are unable to do statistical test such as Cronbach's Alpha for translated tool since the there is no compatible software that accept both local languages. However, the translated languages were back translated to original version and there was no problem of consistency. 

4. Cross check whether in sequential mixed method, you will have to collect data from the same respondents or not and share the reference of mixed-method you have followed. Thank you so much for your valuable and insightful comments. 

Dear respected reviewer, here we tried to remained you that during data collection, we followed the steps which was mentioned here in. To this end, we first collected and analyzed the quantitative data and following this we collected and analyzed qualitative data. The purpose was to use qualitative results to assist in explaining and interpreting the findings of a quantitative study. Hence, we used sequential mixed methods which is termed as explanatory sequential mixed methods. This can be done either by re-contacting appropriate members of the original quantitative sample or by drawing a new sample that meets the purposive criteria. The reference we followed was according to Creswell, 2003 and (Ivankova NV, Creswell JW, Stick SL. Using Mixed-Methods Sequential Explanatory Design: From Theory to Practice. Field Methods. 2006;18(1):3-20). 

End of authors responses for Reviewer 1

Authors’ Response to Reviewer 2’s Comments and Suggestions

Reviewer’s General Comments Overall, thank you very much for your positive and constructive suggestions

1. If the authors have adequately addressed your comments raised in a previous round of review and you feel that this manuscript is now acceptable for publication, you may indicate that here to bypass the “Comments to the Author” section, enter your conflict-of-interest statement in the “Confidential to Editor” section, and submit your "Accept" recommendation.

Reviewer #2: All comments have been addressed

 Thank you very much for your valuable input in advancing this manuscript. 

2. Is the manuscript technically sound, and do the data support the conclusions?

Reviewer #2: Yes Thank you very much!!

3. Has the statistical analysis been performed appropriately and rigorously?

Reviewer #2: Yes Thank you so much. We would like to thank you for your appreciation and constructive suggestion. 

4. Have the authors made all data underlying the findings in their manuscript fully available?

Reviewer #2: Yes Thanks! 

5. Is the manuscript presented in an intelligible fashion and written in standard English?

Reviewer #2: Yes Thank you very much for such valuable intellectual input. 

End of authors responses for Reviewer 2

---

## [Editor Report · Decision Letter 3]

7 Mar 2022

Health status of street children and reasons for being forced to live on the streets in Harar, Eastern Ethiopia. Using mixed methods

PONE-D-21-13888R3

Dear Dr. Eyeberu,

We’re pleased to inform you that your manuscript has been judged scientifically suitable for publication and will be formally accepted for publication once it meets all outstanding technical requirements.

Kind regards,

Faisal Abbas, PhD

Academic Editor

PLOS ONE

Additional Editor Comments (optional):

Accept
---

## [Editor Report · Acceptance letter]

9 Mar 2022

PONE-D-21-13888R3 

*Health status of street children and reasons for being forced to live on the streets in Harar, Eastern Ethiopia. Using mixed methods*

Dear Dr. Eyeberu:

I'm pleased to inform you that your manuscript has been deemed suitable for publication in PLOS ONE. Congratulations! Your manuscript is now with our production department. 

Kind regards, 

on behalf of

Dr. Faisal Abbas 

Academic Editor

PLOS ONE